# Lactoferrin Inhibition of the Complex Formation between ACE2 Receptor and SARS CoV-2 Recognition Binding Domain

**DOI:** 10.3390/ijms23105436

**Published:** 2022-05-13

**Authors:** Roberta Piacentini, Laura Centi, Mattia Miotto, Edoardo Milanetti, Lorenzo Di Rienzo, Martina Pitea, Paolo Piazza, Giancarlo Ruocco, Alberto Boffi, Giacomo Parisi

**Affiliations:** 1Department of Biochemistry, Sapienza University, Piazzale Aldo Moro 5, 00185 Rome, Italy; roberta.piacentini@uniroma1.it (R.P.); centi.laura95@gmail.com (L.C.); alberto.boffi@uniroma1.it (A.B.); 2Center of Life Nano and Neuro Science, Institute of Italian Technology, Viale Regina Elena 291, 00181 Rome, Italy; miottomattia1@gmail.com (M.M.); edoardo.milanetti@uniroma1.it (E.M.); lorenzo.dirienzo@iit.it (L.D.R.); martina.pitea@iit.it (M.P.); giancarlo.ruocco@iit.it (G.R.); 3Department of Physics, Sapienza University, Piazzale Aldo Moro 5, 00185 Rome, Italy; 4D-Tails s.r.l., Via di Torre Rossa 66, 00165 Rome, Italy; 5EDIF Instruments s.r.l., Via Ardeatina 132, 00147 Rome, Italy; paolopiazza@edifinstruments.com

**Keywords:** SARS-CoV-2 receptor-binding domain (RBD), angiotensin-converting enzyme 2 (ACE2), lactoferrin, biolayer interferometry, nanoparticle enhanced turbidimetry, kinetic analysis

## Abstract

The present investigation focuses on the analysis of the interactions among human lactoferrin (LF), SARS-CoV-2 receptor-binding domain (RBD) and human angiotensin-converting enzyme 2 (ACE2) receptor in order to assess possible mutual interactions that could provide a molecular basis of the reported preventative effect of lactoferrin against CoV-2 infection. In particular, kinetic and thermodynamic parameters for the pairwise interactions among the three proteins were measured via two independent techniques, biolayer interferometry and latex nanoparticle-enhanced turbidimetry. The results obtained clearly indicate that LF is able to bind the ACE2 receptor ectodomain with significantly high affinity, whereas no binding to the RBD was observed up to the maximum “physiological” lactoferrin concentration range. Lactoferrin, above 1 µM concentration, thus appears to directly interfere with RBD–ACE2 binding, bringing about a measurable, up to 300-fold increase of the K_D_ value relative to RBD–ACE2 complex formation.

## 1. Introduction

Lactoferrin (LF) is an iron-binding glycoprotein composed of approximately 700 amino acids (molecular weight of about 80 kD, depending on glycosylation extent) folded into two globular lobes connected by an α-helix. LF can bind two ferric ions with a high-affinity metal binding site in each lobe; furthermore, it can also bind Cu^+2^, Zn^+2^ and Mn^+2^ ions. All mammals can produce LF, secreted from cells of the epithelial mucosa within most exocrine fluids, including tears, saliva, nasal and bronchial secretions and, above all, milk, where it is the major iron-binding protein. Lactoferrin has been shown to play a key role in many biological functions related to innate immunity and more in general in the defense against pathogens [1,2,3,4]. In recent years, many experimental observations highlighted novel details within the reported biological activities, with particular focus on the broad antimicrobial action against bacteria, viruses and fungi. Thus, LF biological activities have been attributed only in part to the iron-sequestering activity [5], as they also involve receptor binding, signaling or even protein folding [6,7]. LF has been reported to interact directly with a variety of virus capsid proteins [8], thus suggesting an impairment of viral entry into target cells by blocking the recognition of host cell molecules that the virus itself uses as receptors or co-receptors. Direct binding to viral particles has been described for herpesvirus [9], polio- and rotaviruses [10] and, possibly, human immunodeficiency virus [11] according to mechanisms reviewed by Berlutti et al. [12]. So far, several lines of evidence indicated that a protective role of LF might be also operating towards SARS-CoV-2 coronavirus infection [13,14,15,16,17,18]. In this case, the key cellular receptors of SARS-CoV-2, similarly to those of SARS-CoV and MERS-CoV, have been identified with certainty and have been considered as the main conduit for virus entry within the lower respiratory system [19]. In vivo experiments confirmed that SARS-CoV-2 virus cell entry is mediated by high-affinity interactions between the receptor-binding domain (RBD) of the virus spike glycoprotein and the human host angiotensin-converting enzyme 2 (ACE2) receptor [20]. The spike protein is a 1273-amino acid single-pass transmembrane protein with a short C-terminal tail on the interior of the virus, a transmembrane helix, and a large *N*-terminal ectodomain exposed on the virus external surface. The spike glycoprotein forms homotrimers in which the three subunits interact through their ectodomains. Each subunit contains two topologically and functionally distinct regions named S1 and S2. The S1 regions at the N-terminal end form the portion of the protein furthest from the viral surface within the assembled trimer, whereas the S2 regions form a flexible “stalk” harboring interfaces that hold the trimer in place. High-resolution cryo-EM studies have unveiled the structural determinants that govern the interaction of the spike protein and/or its RBD domain with ACE2, pointing out that the entry of the virus inside the host cells requires the occurrence of a sequence of molecular interactions involving sialic acid and/or heparan sulphate (HF) residues that mediate the attachment of the virion to the cell surface and ultimately lead to spike protein binding to ACE2 receptor and initiate the internalization process [21,22,23,24]. In this framework, the possible molecular mechanisms behind the suggested antiviral action of LF was investigated in a computational study [25,26] based on the search for a possible direct interaction between LF and spike protein or between LF and ACE2 receptor, that could inhibit the RBD/ACE2 complex formation. The results of the computational analysis indicated the presence of possible binding regions having a meaningful shape complementarity for both LF and spike protein and LF and ACE2 receptor. On this basis, the present investigation was dedicated to the direct experimental measurement of the computationally predicted interactions in solution on purified proteins. To this end, the purified ACE2 receptor protein interaction with the RBD domain and with lactoferrin were measured by means of biolayer interferometry (BLI) and latex nanoparticle-enhanced turbidimetry. Turbidimetric assays with latex nanoparticles are widely applied for the detection of biological analytes. The most common applications involve the study of immunoreactivity [27,28] or of the colloidal properties of polymers [29]. In general, biolayer interferometry has been applied in a wide variety of research and development environments for measuring kinetics parameters and for the quantitation of antibodies and proteins, whereas latex nanoparticle-enhanced turbidimetry that exploits molecules adsorption properties to a surface is largely unexplored in non-clinical applications. Nonetheless, it was demonstrated to be a very useful complement to interferometry in order to quantify the interaction between proteins with sufficient reciprocal affinity. Overall, the data indicate that the inhibitory power of lactoferrin on the formation of the ACE2/RBD complex is manifested in its binding to the ACE2 receptor rather than to the RBD domain of the spike protein.

## 2. Results

The ability of LF to bind to specific sites on either the human receptor ACE2 and/or COVID-19 spike’s RBD region was investigated by means of interferometric (BLI) and latex nanoparticle-enhanced turbidimetry measurements. The two independent methodologies were used to assess the affinities and rates of LF binding to ACE2 human receptor and to COVID-19 RBD (see Appendix A for further explanations of the techniques used).

### 2.1. Binding of Lactoferrin to ACE2 and RBD

The molecular interaction of LF with ACE2 and RBD and related and affinities were investigated independently, as depicted in Figure 1. Panels (a) and (b) report the time-courses of the interaction between LF and ACE2, whereas panels (c) and (d) refer to the interaction between LF and RBD.

For the BLI assay (Figure 1a), ACE2 was immobilized on ProA biosensors, and LF was added in 120 s association steps at decreasing concentrations ranging from 33 µM to 0.33 µM, obtained as consecutive dilutions in 1X kinetic buffer. The dissociation rates were measured in a 300 s time range. The time-courses relative to the turbidimetric assay are shown in Figure 1b. Nanospheres with a suitable diameter of 103 nm were coated with ACE2 as described in Materials and Methods. LF was added to the cuvette at concentrations ranging from 21.6 µM to 1.96 nM, and the increase in light absorbance at 340 nm was measured. The same experimental set up was applied to study the binding between LF and RBD, where LF concentration ranged from 10 µM to 50 nM for BLI and from 21.6 µM to 540 nM for turbidimetry. Figure 1c,d show the BLI and turbidimetric analysis for the association of LF to RBD. A clear signal, corresponding to binding or aggregation, was observed between LF and ACE2 ectodomain using both techniques, whereas no signal could be detected, in the same concentration range, between LF and RBD. Table 1 displays the kinetic parameters of LF–ACE2 interaction. The corresponding K_D_ values were 27.64 µM for the interferometric experiment and 46.12 µM in the turbidimetric measurement. These results clearly indicated that LF showed a quantifiable interaction with the ACE2 protein but no binding at all with the RBD protein within the observed concentration range.

### 2.2. Interaction between RBD and ACE2 in the Presence of Lactoferrin

Since LF was effectively demonstrated to bind the ACE2 ectodomain within a µM concentration range whereas no binding to RBD was observed in the same concentration range, the ability of LF to effectively inhibit the interaction between ACE2 and RBD was further investigated by means of interferometric and turbidimetric measurements. The measured affinity between RBD and ACE2 was verified to fall in the nanomolar range, in agreement with reports by Saponaro et al. [30]. In Figure 2a,b, the signals acquired with both BLI and turbidimetric techniques are shown. BLI analysis is presented in Figure 2a and was performed by loading ACE2 on ProA biosensors and allowing the association/dissociation of RBD at concentrations ranging from 1.67 µM to 10 nM. From data fitting analysis performed by BLItz software, the resulting affinity constant was 27.06 nM, in good agreement with previously observed values. The concentrations considered for the analyte protein in the assay were such that one of the proteins was in excess with respect to the other, so the pseudo-first order (PFO) conditions were satisfied. In Figure 2b, aggregation time-courses from turbidimetric assays are shown. The graphs refer to acquisitions of latex nanospheres coated with ACE2 protein and mixed in solution with RBD at decreasing concentrations. The resulting affinity constant was 18.15 nM. Finally, LF was added to a solution with RBD, and the interaction with ACE2 was investigated. In Figure 2c,d, it is possible to observe data acquisition via BLI and turbidimetry of the protein system, where ACE2 was the receptor (on the sensor tip or on the nanoparticles’ surface, respectively), RBD was maintained constant, and LF was present at variable concentrations. The LF effect on the ACE2–RBD complex formation was first investigated by BLI analysis, as shown in Figure 2c. ACE2 was again loaded on ProA biosensors, while RBD was presented to the biosensors in the association step at a fixed concentration of 452.5 nM and then mixed with decreasing concentrations of LF. As a result, both LF and RBD bound to ACE2 fixed to the biosensors, resulting in an apparent K_obs_ in which the binding affinities of both proteins to the receptor were merged. Interestingly, when LF was present in solution at a concentration as low as 45.2 nM, the association signal recorded for the mix was significantly lower than that obtained in the presence of RBD alone, thus suggesting a strong inhibiting effect of LF. The value of K_D_ obtained from the BLItz software was K_D_ = 101.30 nM. The same mixing approach was then applied to the turbidimetric methodology, by mixing in the cuvette nanospheres coated with ACE2 as receptor and a solution at a constant concentration of RBD (19.6 nM) in the presence of decreasing concentrations of LF. An inhibitory effect on the aggregation of ACE2 with RBD was observed, as shown in Figure 2d.

To further analyze the data shown in Figure 2d, curve fitting of the saturation curve (Figure 3a) and of the observed rates (Figure 3b) was executed. According to Equation (S7), the saturation of binding sites expressed in terms of ΔAbs exhibited a hyperbolic dependence on the concentration of the ligand protein, i.e., LF. When expressing protein concentration in logarithmic form, the curve appeared sigmoidal, and the flex point corresponded to an “apparent” dissociation constant K_obs_ of the reaction. In Figure 3a, the values of ΔAbs were plotted against log_10_(LF). The difference in absorbance for each concentration of LF was calculated between the start of step 4 of the turbidimetric assay (see Appendix A) and 600 s after it. In Figure 3b, the initial rates, measured on the same experimental curves, are reported.

### 2.3. Computational Recognition of the Binding Regions of ACE2 to LF

To investigate the possible binding mechanism between ACE2 and LF, the molecular structures of human ACE2 (PDB id: 1R42) and human holo lactoferrin (PDB id: 1LFG) were inspected for portions of the molecular surfaces with high shape complementarity, which can indicate possible binding site candidates (see Methods and [25]). In [26], a computational approach based on the Zernike formalism was applied in order to investigate the possible role of LF in inhibiting SARS-CoV-2 attachment/entrance to the host cells. In particular, the study probed the shape complementarity between Lt and ACE2 focusing on the region where the latter binds to SARS-CoV2 spike protein. Here, we extended the analysis looking for possible bindings between LF and ACE2 receptor considering their whole surfaces. In particular, possible binding between LF and ACE2 receptor was studied, and a set of ACE2 receptor regions with high binding propensity were identified, making these possible candidates. Figure 4a shows the residues of ACE2 having binding propensity scores higher than 0.85 with human LF. There are three regions with marked complementarity. Panel b,c and d of Figure 4 show the identified regions on the molecular surface of human ACE2. As widely discussed in [26], the region with the highest probability of interaction is located in the part of ACE2 that in physiological conditions is difficult to access, as it faces the cell membrane (see Figure 4c). However, there are other regions, which are more exposed, characterized by good shape complementarity with other molecular surface patches of LF that may mediate the interaction (see Figure 4b,d).

## 3. Discussion

In the present work, the effect of human lactoferrin (LF) on the interaction between the SARS-CoV-2 RBD domain and the human ACE2 receptor was investigated in solution by means of two independent experimental methodologies, namely, biolayer interferometry and latex nanoparticle-enhanced turbidimetry. The rationale of this experimental set up was based on previous proposals that suggested possible mechanisms of antiviral action of lactoferrin [13,14,31,32] and on computational studies [26,33,34,35] that revealed significant surface complementarity among lactoferrin and two binding regions on the RBD domain and several possible binding sites to ACE2. Thus far, solution measurements were carried out in order to single out kinetics and thermodynamics parameters for the interaction between lactoferrin and RBD or ACE2, separately. The experimental results in solution pointed out clearly that LF binds to the ACE2 protein ectodomain (K_D_ = 27.6 µM in BLI experiments, K_D_ = 46.1 µM in turbidimetric experiments), whereas no thermodynamically significant interaction could be detected for LF binding to RBD.

In a second set of experiments, the inhibitory effect of lactoferrin on the complex formation between RBD and ACE2 was investigated, starting from a reassessment of RDB–ACE2 complex formation by means of interferometry and turbidimetry methods. The reported K_D_ values for ACE2–RBD interaction, estimated by either BLI or Surface Plasmon Resonance (SPR), ranged between 1.2 nM to 133.3 nM, as reviewed by Saponaro et al. [30]. In the current experimental set up, at 25 °C and in PBS buffer pH 7.4, the calculated K_D_ values were 27.06 nM (BLI) and 18.15 nM (turbidimetry), in good agreement with reported measurements. Thereafter, the effect of increasing concentrations of LF on the formation of the RBD–ACE2 complex was investigated in experiments in which ACE2 was bound either to the biosensor surface (BLI measurements) or to the latex nanoparticle surface (turbidimetric measurements). Overall, the body of the in vitro experimental results thus obtained indicated that LF is indeed capable of inhibiting RBD–ACE2 interaction in solution at concentrations close to the physiological one in human milk (1–10 µM). In particular, as shown in Figure 2 and Figure 3, the apparent dissociation constant (K_obs_) of the RBD–ACE2 complex increased up to 300-fold as a function of LF concentration in solution. A full thermodynamic model for the complex ternary interactions, however, could not be worked out at the present stage, and the values of K_obs_ rest on a phenomenological basis.

The overall picture emerging from the present investigation is that human lactoferrin can be active in directly impairing spike protein binding to the ACE2 receptor and thus limit viral entry into epithelial cells. On the basis of previous computational studies, a possible structural model for ACE2–LF interaction was proposed, as depicted in Figure 4. The model suggested that the region of binding of LF does not strictly overlap with the well-known ACE2–RBD binding interface. As such, two different inhibition scenarios can be hypothesized: on one side, LF binding on an ACE2 surface region far from the interaction surface with RBD (such as in Figure 4c) can lead to structural rearrangements on the ACE2 ectodomain that could effectively prevent spike attachment to host cells; on the other hand, LF attachment on regions highlighted in Figure 4b,d could hamper directly ACE2–RBD interaction. Altogether, our results vouch for a future deeper investigation on the possible bound conformations and their relative stabilities. In fact, the found binding propensities may help guide docking algorithms and subsequent tailored molecular dynamics simulations [36,37,38] which could identify the effective binding regions. As an additional remark, BLI and nanoparticle-enhanced turbidimetry proved to be reliable, independent and efficient techniques for the detection and analysis of molecular interactions and binding; hopefully, the present work can be considered for future applications of these techniques.

## 4. Materials and Methods

### 4.1. Kinetic Analysis of the LF Inhibition Effect on the ACE2–RBD Complex

Human milk lactoferrin (LF) protein was from Sigma-Aldrich, with molecular weight between 82.4 and 87 kD; the estimated extinction coefficient was ε_mM_ = 110.96 mM^−1^ cm^−1^ (280 nm), and the pI was 8.7. The storage buffer was 10 mM phosphate buffered saline at pH 7.4; therefore, all dilutions were executed in the same buffer.

The wild-type receptor-binding domain (RBD) of SARS-CoV-2 spike protein was provided by GenScript. The protein is a C-term HIS-tagged recombinant protein with a predicted molecular weight of 30 kD. The storage buffer was phosphate-buffered saline at pH 7.2, and the calculated pI was 8.91; the stock concentration was 0.89 mg/mL according to the manufacturer. The angiotensin-converting enzyme 2 (ACE2) was also provided by GenScript. The ACE2 ectodomain is a C-term Fc-tagged recombinant protein with predicted molecular weight of 110.5 kD at a concentration of 1.34 mg/mL, in a storage buffer consisting of 20 mM Tris-HCl, 300 mM NaCl, 1 mM ZnCl_2_ and 10% glycerol, pH 7.4. The protein pI is 5.49.

Biolayer interferometry is a label-free technology for measuring biomolecular interactions. It is an optical analytical technique that analyzes the interference pattern of white light reflected from two surfaces: a layer of immobilized protein and an internal reference layer. The binding between a ligand immobilized on the biosensor tip surface and an analyte in solution produces an increase in optical thickness at the biosensor tip, which results in a shift in the interference pattern measured in nanometers. This interaction was measured in real time. BLI has been widely used in the study of biomolecular antibody–antigen interactions [39,40,41], with particular focus, recently, on SARS-CoV-2 antibodies investigation [42].

BLI assays were performed by means of the BLItz system (Sartorius). A description of a BLI assay is presented in Appendix A. Protein-A (ProA), anti-His tag (HIS2) and nickel nitriloacetic acid (Ni-NTA) were selected as biosensors and provided by Sartorius. ACE2 Fc-tagged was immobilized on ProA biosensors, with no aspecific binding to all other experimental components, whereas RBD His-tagged was immobilized on either HIS2 or NTA biosensors, the latter being more susceptible to aspecific binding of non-his-tagged proteins. The biosensors were first equilibrated 10 min in 1X kinetic buffer (Sartorius) consisting of PBS with 0.02% Tween20, 0.1% BSA and 0.05% NaN_3_. Afterwards, depending on the assay, either ACE2 or RBD was loaded on the corresponding biosensor at a concentration of 50 µg/mL (corresponding to C_ACE2_ = 452.5 nM and C_RBD_ = 1.67 µM), as indicated in Sartorius biosensors’ datasheets, for an appropriate time interval. To reach the maximum binding capacity in every experiment, the time length of each experimental step was tested, and each experiment was designed accordingly. The concentration range for each associating protein was chosen, when possible, based on the K_D_ value available from the literature or experimentally determined for every different scenario where K_D_ values were unknown. The data recorded were analyzed by means of BLItz software and MATLAB to extrapolate the kinetic parameters. All association and dissociation curves were fitted by a single exponential function. A description of the instrument and of the analytical model of the system is reported in the Appendix A. Pseudo-first order (PFO) conditions are met when the initial concentration of one of the two reagents is in large excess with respect to the other (between 50- and 100-fold) [43,44,45]. Each acquisition was repeated twice to confirm reproducibility.

The method for the turbidimetric assay consisted in performing a time-course acquisition of light absorption signals with a spectrophotometer (Jasco V-750 UV-Visible/NIR) at fixed wavelength (340 nm) and bandwidth (10 nm) for 1300 s. For the turbidimetric assays, chloromethylated (-CH_2_-Cl) polystyrene nanospheres (furnished by Ikerlat) were used, with a diameter of 103 nm according to the manufacturer. The nanosphere stock concentration was 0.1 g/mL (10%). The parking area was 95 Å^2^/group. Before the assay, the nanospheres were coated with the receptor protein in 10 mM phosphate buffer (PB) at pH 7.4 in the case of ACE2 protein. A total of 2 mL of the solution of ACE2-coated nanospheres was obtained by adding 4 µL of Ikerlat nanospheres to 1.92 mL of PB. Such an amount of nanospheres guarantees 0.02% of solids in the final solution. Then, 26 µL of ACE2 (stock concentration 1.34 mg/mL) was added to the nanospheres solution (final ACE2 concentration 17.42 µg/mL = 157.65 nM), and the solution was kept in gentle agitation on a tilting platform at room temperature for 3 h. Finally, 52 µL of blocking buffer was added. Such buffer consisted of goat serum (Millipore) at a concentration of 20 mL/L and ProClin 300 biocide (Sigma-Aldrich) at 0.3 mL/L in PBS at pH 7.4, to fully block potentially unbound chloromethyl groups. This reaction was incubated again on a tilting surface, gently mixing overnight. In the case of the RBD protein, the coating buffer was 20 mM sodium carbonate at pH 9.2 containing 150 mM NaCl. For the coating step of nanospheres with RBD as the receptor protein, the procedure was the same but with different quantities: 1 µL of Ikerlat nanospheres was added to 0.5 mL of bicarbonate and guaranteed 0.02% of solids in the final solution (the stock concentration was 0.1 g/mL). The concentration of RBD used for the coating procedure was 174.2 µg/mL = 5.8 µM.

For spectra acquisition, polymethyl methacrylate (PMMA) cuvettes with 500 µL capacity and 0.6 cm optical path were used. Appendix A shows an example of data acquisition in the turbidimetric assays performed. Nanoparticle aggregation results in an increasing signal of absorbance attributed to RBD–ACE2 complex formation and was analyzed in phenomenological terms as a saturation curve (see Appendix A for a brief discussion). Thus, a turbidimetric assay was performed with nanospheres coated with fixed concentrations of either ACE2 or RBD, while LF was present at different concentrations. The analysis was carried out in phosphate-buffered saline at pH 7.25 with 0.1% glycine, 0.1% NaN_3_ and polyethylene glycol (PEG) 6000 at a concentration of 5%. The analysis of the turbidimetric data was performed by means of a MATLAB custom program.

### 4.2. Computational Recognition of Binding Regions of ACE2 to LF

The computational analysis was performed starting from the crystallographic structures of human lactoferrin in the holo form (PDB id: 1LFG) and that of human ACE2 (PDB id: 1R42). For both protein structures, we used DMS [46] to compute the solvent-accessible surface, using a density of 5 points per Å^2^ and a water probe radius of 1.4 Å. The unit normal vector, for each point of the surface, was calculated using the flag −n. The DMS software returns a discretized version of the molecular surface, that is represented by a set of points in the three-dimensional space. To find possible interacting regions, we looked for regions of molecular surfaces with high shape complementarity. Operatively, we compared all possible regions of ACE2 with all possible regions of LF. We defined surface regions, i.e., patches, as the group of points that fell within a sphere of radius R s = 6 Å, centered on each of the points of the surfaces. Once the patch was selected, we described it on the basis of the Zernike polynomials [26], i.e., we associated to each patch a set of coefficients that represented the patch in the Zernike basis. Once a patch was represented in terms of its Zernike descriptors, the complementarity between that patch and another one could be simply measured as the Euclidean distance between the invariant vectors. We thus associated to each point of the two surfaces the minimum distance value observed—the binding propensity—between the considered point and all points of the other surface. After all surface points were associated with their binding propensity, we performed a smoothing process to highlight signals in specific regions characterized mostly by low-distance values. In this process, each point was associated with the mean value of the points in its neighborhood: the basic idea is that the interacting region should be mostly made up of elements with high complementarity and, therefore, a high average value of binding propensity (see [26] for further details). For both the patch definition and the smoothing process, we adopted a sphere radius of 6 Å.

## 5. Conclusions

In the present work, the inhibiting effect of human lactoferrin on the complex formation between the spike protein binding domain RBD and the ACE2 receptor, was demonstrated directly for the first time. In detail, LF showed a remarkable binding propensity towards the ACE2 receptor, whereas no affinity to RBD was established. Moreover, when LF was mixed at a concentration of 1µM with RBD, a strong decrease in binding affinity between ACE2 and wild type RBD was observed, thus suggesting the ability of LF to inhibit, in vitro, the formation of such complex. Three distinct binding sites were computationally identified for ACE2–LF interaction, enlightening two different inhibition mechanisms through which lactoferrin could exert its inhibiting properties. The detailed mechanism of LF inhibition of the complex formation remains unclear at the moment. Considering the present lack of a preventative regimen established for COVID-19, the use of LF as a local protection might be increasingly desirable as an adjunct protection against SARS-CoV-2 infection.

## Figures and Tables

**Figure 1 ijms-23-05436-f001:**
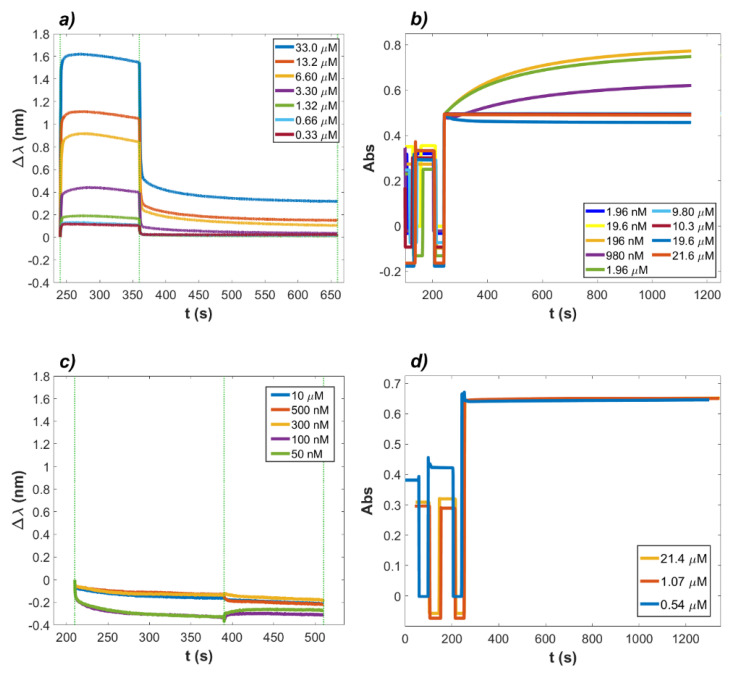
Time−courses of the binding between LF and ACE2 and LF and RBD. (**a**) BLI signals of LF as the analyte protein at different concentrations with ACE2 loaded on the biosensor (ProA). The vertical dashed lines indicate the duration of the binding step (120 s) and of the dissociation step (300 s). The value of K_D_ obtained from the BLItz software was K_D_ = 27.64 µM. (**b**) Turbidimetry assay performed with LF as the analyte protein at variable concentrations in solution with latex nanospheres coated with ACE2. From the data analysis performed using Equations (S3)–(S5), the value of K_D_ obtained was 46.12 µM. (**c**) BLI signals of LF as the analyte protein at different concentrations with RBD loaded on the biosensor (HIS2). The binding step was 180 s, the dissociation step was 120 s. (**d**) Turbidimetry assay performed with LF as the analyte protein at variable concentrations in solution with latex nanospheres coated with RBD. No association of LF to RBD was recorded in both cases.

**Figure 2 ijms-23-05436-f002:**
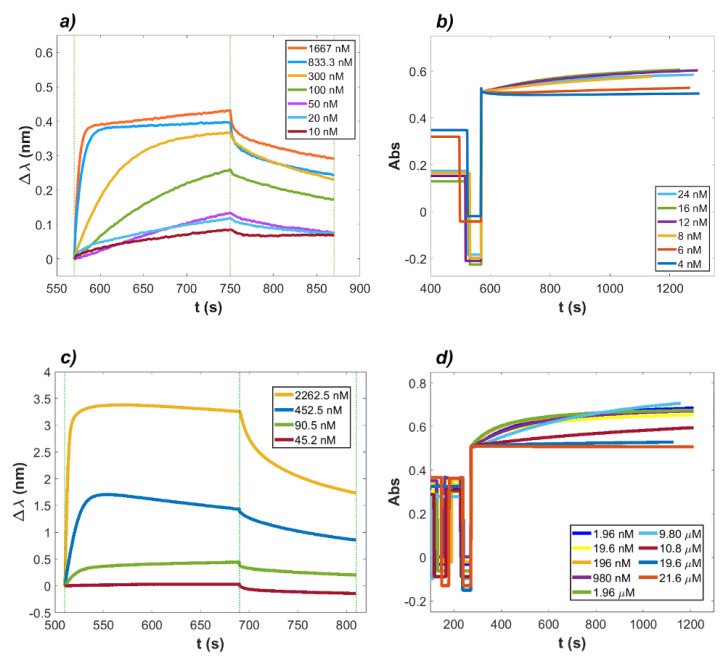
Time−courses of the reaction between RBD in solution with ACE2 in the absence (panels a and b) and in the presence (panels c and d) of human lactoferrin (LF). (**a**) Signals of the binding and dissociation experiment performed via BLI with the protein system RBD and ACE2. ACE2 was loaded on ProA biosensors, and RBD was in solution at decreasing concentrations. The vertical dashed lines indicate the time interval of the binding step (180 s) and of the dissociation step (120 s). (**b**) Aggregation signals from turbidimetric assays. Latex nanospheres were coated with the ACE2 protein and mixed in solution with RBD at decreasing concentrations. Panels (**c**,**d**): Time-courses of the reaction with RBD and ACE2 in the presence of LF in solution. (**c**) BLI signals of LF in solution with RBD as the analyte protein at different LF concentrations, with ACE2 loaded on ProA biosensors. The vertical dashed lines indicate the start of the binding step (180 s) and of the dissociation step (120 s). (**d**) Turbidimetry assay performed with LF at variable concentrations and RBD present at a fixed concentration, in solution with latex nanospheres coated with ACE2.

**Figure 3 ijms-23-05436-f003:**
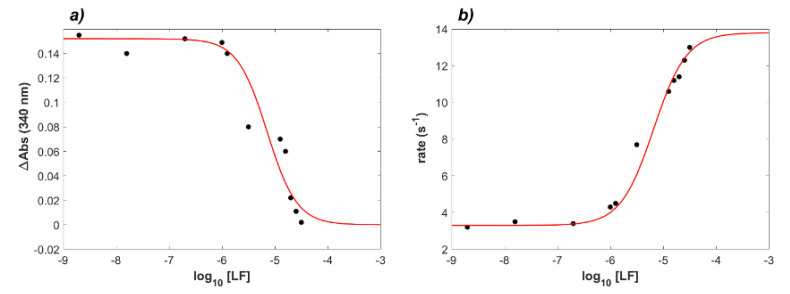
Inhibition of RBD−ACE2 complex formation in the presence of increasing concentrations of lactoferrin. The turbidimetric data of Figure 2d were used to analyze the effect of increasing lactoferrin concentration (in log scale). The absorbance amplitudes of the reactions are shown in panel (**a**), whereas the initial rates of the same curves are depicted in panel (**b**). Interpolating red curves represent the best fit to the data obtained by Equation (S8). The data analysis, performed via custom MATLAB program, yielded apparent K_obs_ values of 9.5 ± 1.5 µM (**a**) and 6.3 ± 1.2 µM (**b**), respectively, with cooperativity coefficient n of 1.44 ± 0.11 and 1.54 ± 0.08, respectively.

**Figure 4 ijms-23-05436-f004:**
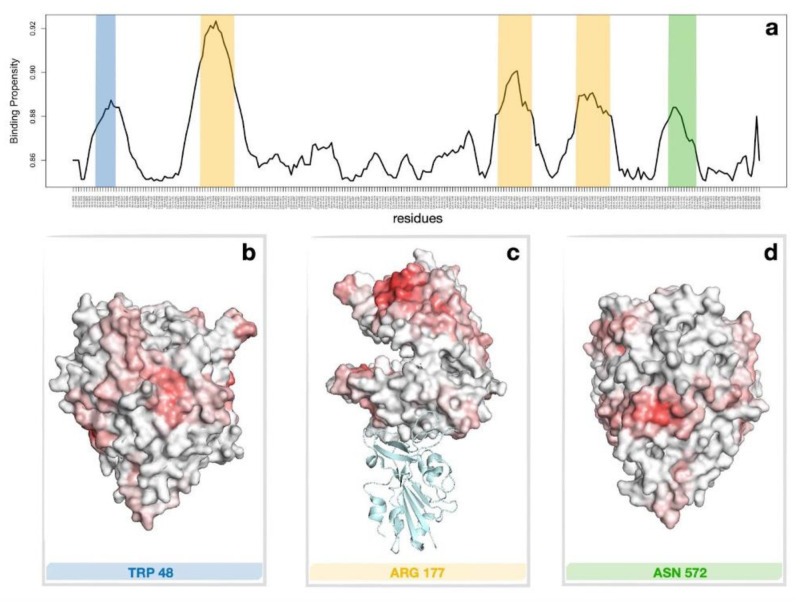
Regions of high binding propensity of ACE2 to LF. (**a**) Binding propensity of human ACE2 residues for human lactoferrin obtained on the basis of local shape complementarity of the molecular surfaces [7]. Only residues whose binding propensity is higher than 0.85 are reported. Blue, orange, and green bands highlight three different portions of the molecular surfaces characterized by high binding propensity. (**b**) Molecular surface of the extracellular region of human ACE2 colored according to the Zernike binding propensity score. The color turns from white to red as the local binding propensity increases. The surface was oriented to show the region around residue Tpr48, which is comprised in the blue band in panel (**a**). (**c**) Same as in (**b**) but displaying the region around residue Arg177 and marked with the orange bands; a cartoon representation of the RDB of the spike protein bound to ACE2 is also shown. (**d**) Same as in (**b**), but for the green band of the panel (**a**) marking the region around residue Asn572.

**Table 1 ijms-23-05436-t001:** Kinetic parameters relative to LF and ACE2 interaction. The k_on_ and k_off_ parameters were estimated by single exponential fit of the curves (see Appendix A), and the corresponding K_D_ values were calculated according to Appendix A.

	k_on_ (M^−1^ s^−1^)	k_off_ (s^−1^)	K_D_ (µM)
**INTERFEROMETRY**	(166.90 ± 4.79) 10^2^	0.461 ± 0.007	27.64 ± 0.91
**TURBIDIMETRY**	33.41 ± 5.41	(1.54 ± 0.39) 10^−3^	46.12 ± 12.12

## Data Availability

Not applicable.

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
