# Peer review of "Lactoferrin Inhibition of the Complex Formation between ACE2 Receptor and SARS CoV-2 Recognition Binding Domain"

_ijms, 2022, doi:10.3390/ijms23105436_

Round 1
Reviewer 1 Report
The work by Piacentini et al about the inhibition role of Lactoferrin on the ACE2/RBD complexation of SARS-CoV-2 both in vitro and in silico shows conceptual problems which must be addressed before publication. See below major comments:
1) I believe the authors should also comment on the results by Koelher et al. (https://doi.org/10.1038/s41467-021-27325-1) which describe the ACE2/RBD binding constant for several VoCs. For instance, mutations are simply decreasing the affinity constant (~1/Kd) with ACE2 receptor, with Kd as the equilibrium dissociation constants between RBD of VoCs and ACE2. In the same paper, the KD are reported for each VoC case with absolute values of KD in the range of 21 (gamma ) to 134 (WT) nM.
The variant used in this study must be clear out, I guess due to the reported KD ~ 101.30 nM with no inference about the error bars, it shall correspond to WT.
2) It is not clear for me, how having a very weak KD in order of μM for Lactoferrin and ACE2, the authors can claim a pesudo inhibition of the ACE2/RBD, as the main result of the MS. The latter interaction is 3 orders of magnitude larger (see anove), meaning a high strength of recognition. In the time scale of recognition, first ACE2/RBD will be formed rapidly and then slow binding of ACE2/Lactoferrin will occur if any biomolecule is left.
3) Finally, the MD part is completely missing a proper computational method section. No detailed information for reproduction of results about the method and technical analysis of the docking protocol ACE2/Lactoferrin.
4) I believe also, MD relaxation of the complex ACE2/Lactoferrin should be provided and analysis of the interaction at the protein-protein interface using contact maps (see doi.org/10.1021/acs.jcim.9b00883 or doi.org/10.3389/fmolb.2021.619381), event at static maner should be provided.
Author Response
REVIEWER 1
The work by Piacentini et al about the inhibition role of Lactoferrin on the ACE2/RBD complexation of SARS-CoV-2 both in vitro and in silico shows conceptual problems which must be addressed before publication. See below major comments:
1) I believe the authors should also comment on the results by Koelher et al. (https://doi.org/10.1038/s41467-021-27325-1) which describe the ACE2/RBD binding constant for several VoCs. For instance, mutations are simply decreasing the affinity constant (~1/Kd) with ACE2 receptor, with Kd as the equilibrium dissociation constants between RBD of VoCs and ACE2. In the same paper, the KD are reported for each VoC case with absolute values of KD in the range of 21 (gamma ) to 134 (WT) nM. The variant used in this study must be clear out, I guess due to the reported KD ~ 101.30 nM with no inference about the error bars, it shall correspond to WT.
Our analysis was conducted on a single variant (corresponding to Wild Type) and does not take into account mutations from other VoC’s. Since the KD reported in literature for the WT seems to have a relevant variability, also depending on the specific technique used for measurements (see Saponaro et al., in which WT constructs of different length, showed a KD ranging 3 to 130nM) or a pretty wide error range (Koelher et al.), our result of 27.06 ± 0.63 nM falls in good agreement with data available in literature. The KD 101.3 ± 2.4 nM is coming from data obtained in presence of lactoferrin in solution with RBD, corresponding to a 3-fold decrease in binding capacity. Nevertheless, a research focused on the variability of the effect of LF on VoC’s could be definitely desirable. On the other hand, the manuscript will be corrected to specify the nature of the RBD used for our purposes.
2) It is not clear for me how having a very weak KD in order of μM for Lactoferrin and ACE2, the authors can claim a pesudo inhibition of the ACE2/RBD, as the main result of the MS. The latter interaction is 3 orders of magnitude larger (see anove), meaning a high strength of recognition. In the time scale of recognition, first ACE2/RBD will be formed rapidly and then slow binding of ACE2/Lactoferrin will occur if any biomolecule is left.
On this insightful comment from the reviewer, I would like to respond taking into account two considerations. When we are in the presence of a trimolecular system, such as that composed of RBD, ACE2 and LF, the experimental signal related to the binding between the different species may result in a single exponential or a multiple exponential taking into account binding of species with different affinities. Considering the inability of LF to bind to RBD at the experimental concentrations used, we can assume the only binding events taking place are those between LF and ACE2, and between ACE2 and RBD. Since the KD for the latter binding event is known to be measurable and fast (27.06nM), in the event of the quick binding of RBD, followed by a slow binding of LF to ACE2 still available on the tip of the biosensors, we would expect a double exponential behavior or the curves. Taking into consideration the R2 of the BLI curves we have a 0.99 for single exponential fitting, suggesting a good agreement between fit and curves. Moreover, as reported in Supplementary data, when we associate RBD to NTA biosensors and use LF (at a constant concentration of 1µM) in mix with decreasing concentrations of ACE2, we witnessed a 2-fold decrease in Kd for the RBD-ACE2 binding. In this case there is no possibility for LF to bind directly to the biosensors since no KD was observed for the RBD-LF interaction (for the working concentrations of LF). The specific interaction mechanism between LF and ACE2 remains still unclear at the moment.
3,4) Finally, the MD part is completely missing a proper computational method section. No detailed information for reproduction of results about the method and technical analysis of the docking protocol ACE2/Lactoferrin.
I believe also, MD relaxation of the complex ACE2/Lactoferrin should be provided and analysis of the interaction at the protein-protein interface using contact maps (see doi.org/10.1021/acs.jcim.9b00883 or doi.org/10.3389/fmolb.2021.619381), event at static manner should be provided.
We see the Reviewer point. In the former version of the manuscript, we opted not to include an extended version of the computational methods section as it mainly overlaps with the one already presented in the reference article [Miotto et al. Front. Mol. Biosci., 15]. However, we agree that this may impede the fast comprehension of the present article. Thus we have now extended the computational method section to explain the adopted procedure. Finally, we would like to specify that no MD simulations were conducted in the present manuscript. The Zernike algorithm is able to assess the binding propensities of regions of a protein to bind with a partner starting from the experimental structure in PDB form, without the need of initial docking procedure to get the putative complex. We believe that the docking and subsequent MD simulation of the LF-ACE2 complex together with contact maps, the Reviewer suggested, could be a direct continuation of this work (whose scope was an experimental investigation of the binding). We have now extended the discussion adding some comments on this future direction, where we pointed to the papers provided by the Reviewer.
Reviewer 2 Report
In the presented for review article from Parisi et al. - " Lactoferrin inhibition of the complex formation between ACE2 2 receptor and SARS CoV2 recognition binding domain" focused the attention of the reader on the analysis of the interactions of lactoferrin (LF) and SARS-CoV-2 receptor-binding domain (RBD) and human Angiotensin-converting enzyme 2 (ACE2) receptor.
The effect of mutual interactions was explored on a molecular basis for the effect of lactoferrin against CoV-2 infection.
With regard to the study comments are as follow:
Abstract:
The assessment of the kinetic and thermodynamic parameters for the pairwise interactions among the three proteins have been measured in parallel by means of biolayer interferometry and latex nanoparticle enhanced turbidimetry.
it is not clear to the reader, why the measurements of the stated parameters are emphasized as a parallel measurement. Explained?
Lactoferrin, above 1 μM concentration, thus appears to directly interfere with the RBD-ACE2 binding, bringing about a measurable, up to a 300-fold increase of the KD value relative to the RBD-ACE2 complex formation.
What was the tested conetration range for claiming the results?
Introduction part:
The intro part was not well stated. The citted literature for the molecular mechanisms behind the suggested antiviral action in a computational frame citted [23, 24] searching for a possible direct interaction between LF and Spike protein or between LF and ACE2 receptor, was not enough and supporting the hypothesisie only with the citted literature.
The lack of the cittetion of the papers stating the capacity of the computationa methods for the widly explored in the last two years proble. The computation methods were explored a lot in that direction.
The authors neglected a bit the fforts in the filed. Recommned for cittetion: https://doi.org/10.3390/ph14121328 https://doi.org/10.3390/molecules26082157
The sectins with results and discussion:
About the Figure 4. Regions of high binding propensity of ACE2 to LF.
The choosen representation was not cleare with the stated message from the authors.
Most of the presented results reported that the residues of ACE2 having binding scores higher than 0.85 with human L are based on the literature data. The lack of the own approach from the autors for me is a vety weka point. The docking study should be conducted.
The concluding part was a scasly presented. Should be reformulated and presented in more clear form. A pattern between the results and explanation was not revealed or the message was not succifintly well explosed for the reader.
Author Response
1) Abstract: The assessment of the kinetic and thermodynamic parameters for the pairwise interactions among the three proteins have been measured in parallel by means of biolayer interferometry and latex nanoparticle enhanced turbidimetry. it is not clear to the reader, why the measurements of the stated parameters are emphasized as a parallel measurement. Explained?
We refer to the two different techniques used as “in parallel” since the experimental setting was designed to determine the same kinetic parameters making use of the same protein constructs (RBD, ACE2 and LF). In accordance with the reviewer perplexity we decided to remove the statement “in parallel” from the manuscript.
2) Lactoferrin, above 1 μM concentration, thus appears to directly interfere with the RBD-ACE2 binding, bringing about a measurable, up to a 300-fold increase of the KD value relative to the RBD-ACE2 complex formation. What was the tested conetration range for claiming the results?
In regard to “Lactoferrin, above 1 μM concentration, thus appears to directly interfere with the RBD-ACE2 binding, bringing about a measurable, up to a 300-fold increase of the KD value relative to the RBD-ACE2 complex formation.”, the 300 fold increase is referred to the results obtained through the turbidimetric method. In details, we witnessed a decrease of Kd value from 18nM for the binding between WT-RBD and ACE2 to 45µM when the same experiment was conducted in the presence of increasing concentrations of LF, corresponding to an approximately 300-fold decrease in relative affinities between ACE2 and WT-RBD. The concentration range of LF used in the experiment was 1.96nM to 21.6µM, as reported in the legend of figure 2.d.
3) Introduction part: The intro part was not well stated. The cited literature for the molecular mechanisms behind the suggested antiviral action in a computational frame cited [23, 24] searching for a possible direct interaction between LF and Spike protein or between LF and ACE2 receptor, was not enough and supporting the hypothesisie only with the cited literature. The lack of the citation of the papers stating the capacity of the computational methods for the widly explored in the last two years proble. The computation methods were explored a lot in that direction. The authors neglected a bit the efforts in the field. Recommend for citation: https://doi.org/10.3390/ph14121328 https://doi.org/10.3390/molecules26082157
In consideration of the intro part, the motivation for this research comes from findings (both computational and in vitro) reported in the cited literature from reference 8 to 24. The papers referred as 23 and 24 report previous findings from our same group, the in vitro results of the present research was intended to confirm experimentally the results obtained computationally in 23 and 24. We agree with the reviewer that a deeper commitment in literature citing could be beneficial to frame the research, therefore citations suggested are going to be included in the manuscript.
4) The sections with results and discussion: About Figure 4. Regions of high binding propensity of ACE2 to LF. The chosen representation was not cleared with the stated message from the authors. Most of the presented results reported that the residues of ACE2 having binding scores higher than 0.85 with human L are based on the literature data. The lack of the own approach from the authors for me is a very weak point. The docking study should be conducted.
We see the Reviewer point. In the previous version of the manuscript the computational analysis summarized in Figure 4 was poorly described and the novelty with respect to the reference article [Miotto et al. Front. Mol. Biosci., 15] was not made sufficiently clear. While we opted not to include an extended version of the computational methods section as they mainly overlap with those already presented in [Miotto et al. Front. Mol. Biosci., 15], the analysis of the ACE2-Lt binding propensities is a novel result, not presented/discussed in literature. We have now extended both the Result and Methods sections specifying the novelties with respect to Miotto et al. Front. Mol. Biosci., 15. Moreover, we would like to stress that the Zernike algorithm is able to assess the binding propensities of regions of a protein to bind with a partner starting from the experimental structure in PDB form, without the need of initial docking procedure to get the putative complex. We believe that the docking (and subsequent MD simulation, suggested by Reviewer 1) of the LF-ACE2 complex could be a direct continuation of this work (whose scope was an experimental investigation of the binding). We have now also extended the discussion adding some comments on this future direction.
- The concluding part was scarcely presented. Should be reformulated and presented in a more clear form. A pattern between the results and explanation was not revealed or the message was not sufficiently well exposed for the reader.
We see the review point and we agreed a deeper detail was needed in the conclusions in order to get the message of this research more clear and to the point. We modified the main text of the manuscript accordingly.
Round 2
Reviewer 1 Report
I am glad to endorse the MS for publication and appreciate the discussion on kinetic of three components by the authors.
Reviewer 2 Report
The authors did the required corrections.